# Intervertebral Disc Regeneration Injection of a Cell-Loaded Collagen Hydrogel in a Sheep Model

**DOI:** 10.3390/ijms22084248

**Published:** 2021-04-19

**Authors:** Andrea Friedmann, Andre Baertel, Christine Schmitt, Christopher Ludtka, Javorina Milosevic, Hans-Joerg Meisel, Felix Goehre, Stefan Schwan

**Affiliations:** 1Department of Biological and Macromolecular Materials, Fraunhofer Institute for Microstructure of Materials and Systems IMWS, 06120 Halle, Germany; andrea.friedmann@imws.fraunhofer.de (A.F.); christine.schmitt.paracelsus@web.de (C.S.); 2Department of Veterinary Medicine, University of Leipzig, 04103 Leipzig, Germany; andre.baerthel@web.de; 3Department for Orthopaedics and Traumatology, Martin Luther University, Halle Wittenberg, 06120 Halle, Germany; 4Department of Biomedical Engineering, University of Florida, Gainesville, FL 32611, USA; cludtka@ufl.edu; 5Spinplant GmbH, 06120 Halle, Germany; javorina.milosevic@spinplant.de; 6Department of Neurosurgery, BG Klinikum Bergmannstrost, 06110 Halle, Germany; hans-joerg.meisel@bergmannstrost.de (H.-J.M.); felix.goehre@bergmannstrost.de (F.G.); 7Department of Neurosurgery, University of Helsinki and Helsinki University Hospital, 00260 Helsinki, Finland

**Keywords:** adipose stem cell, collagen hydrogel, intervertebral disc regeneration, injection, ovine model, regenerative medicine

## Abstract

Degenerated intervertebral discs (IVDs) were treated with autologous adipose-derived stem cells (ASC) loaded into an injectable collagen scaffold in a sheep model to investigate the implant’s therapeutic potential regarding the progression of degeneration of previously damaged discs. In this study, 18 merino sheep were subjected to a 3-step minimally invasive injury and treatment model, which consisted of surgically induced disc degeneration, treatment of IVDs with an ASC-loaded collagen hydrogel 6 weeks post-operatively, and assessment of the implant’s influence on degenerative tissue changes after 6 and 12 months of grazing. Autologous ASCs were extracted from subcutaneous adipose tissue and cultivated in vitro. At the end of the experiment, disc heights were determined by µ-CT measurements and morphological tissue changes were histologically examined.Histological investigations show that, after treatment with the ASC-loaded collagen hydrogel implant, degeneration-specific features were observed less frequently. Quantitative studies of the degree of degeneration did not demonstrate a significant influence on potential tissue regeneration with treatment. Regarding disc height analysis, at both 6 and 12 months after treatment with the ASC-loaded collagen hydrogel implant a stabilization of the disc height can be seen. A complete restoration of the intervertebral disc heights however could not be achieved.The reported injection procedure describes in a preclinical model a translational therapeutic approach for degenerative disc diseases based on adipose-derived stem cells in a collagen hydrogel scaffold. Further investigations are planned with the use of a different injectable scaffold material using the same test model.

## 1. Introduction

Degenerative disc disease is a clinical syndrome, which occurs through permanent structural changes of the intervertebral disc, and is expressed clinically by its associated symptomatology. Through advancing epidemiological changes at the population level, disc degeneration and its secondary diseases will continue to increase in importance in the future. Currently, numerous nonsurgical, as well as surgical treatment methods are available, which are mainly used for the relief of symptoms and a slowdown of the degenerative changes. However, these interventions are limited because they do not restore native disc structure or function. Treatment may even exacerbate degeneration of adjacent healthy discs [1].

Regenerative implants based on autologous cells or cell structures can offer an alternative [2]. For the extraction of autologous cells for intervertebral disc therapies, different options have been previously reported. On one hand, the direct removal of cells and matrix from the degenerated intervertebral disc itself is possible [3,4]. Further, the extraction of allogeneic cells or even complete intervertebral discs from donors has been described, though of course with consideration for potential immunological rejection [5]. However, the extraction of a sufficient number of cells is often not possible, especially if the available tissue demonstrates severe degradation. The dedifferentiation of these cells in culture also restricts the possibilities of cultivation [6,7]. Implants based on mesenchymal stem cells (MSC) offer an alternative to avoid these limitations. The MSCs are harvested from the mesenchyme, then expanded in vitro and rearranged in 3D cultures mimicking cell-cell and cell-matrix interactions. From both clinical and ethical points of view, the extraction of these cells, which constitute only 0.001 to 0.01 % of cells in the bone marrow, is questionable due to the collateral tissue damage involved in their extraction [8,9]. An alternative could be the use of adipose-derived stem cells (ASC), which are obtained from fat tissue. They are considered to have comparable potential to bone marrow-derived stem cells, which in vitro investigations appear to confirm [10]. Their extraction is ethically unobjectionable and facilitates the transfer of autologous ASCs within a single surgery. ASCs demonstrate the capacity for pluripotent mesenchymal differentiation and can be easily expanded under standard tissue culture conditions. Their therapeutic effect has been evaluated by in vitro culture models, as well as in vivo small and large animal models [11]. In particular, large animal models are of special interest regarding tissue comparability when considering applications in humans. However, to develop and test the potential of new therapeutic approaches, it is necessary to comprehensively simulate the inherent complexity of the intervertebral disc, with biochemical, biomechanical, nutritional, and metabolic factors at play. Therefore, an ovine animal model has been established for replicating disc degeneration via a far lateral, minimally invasive nucleotomy [12]. It is characterized by a proven comparability to human intervertebral disc anatomy, intradiscal pressure, biomechanical stress, and cell physiology [13,14,15]. In fact, there are several approaches for disc treatment aiming to provide both long-term regeneration of the disc structure as well as restoring mechanical function. These strategies range from intradiscal injection of stem cells or growth factors to hydrogel development for nucleus pulposus replacement as well as total disc replacement by tissue engineered grafts [16,17,18]. A great number of competing material systems appear suitable for in vitro investigations. Due to anatomical restrictions regarding access to the nucleus region of the intervertebral disc, injectable scaffold materials are of particular interest from a clinical point of view, as they can be injected as a liquid together with cells directly into the nucleus area in a minimally invasive way followed by rapid in situ gelation [16].

Since IVD degeneration is primarily due to the breakdown of matrix proteins such as collagen, the development of effective treatment strategies should focus on the regeneration of the ECM in addition to the cellular components. ECM-mimetic biomaterials for the efficient treatment of IVD degeneration are a promising therapeutic approach [19]. The use of collagen as an injectable biomaterial for the regeneration of both annulus fibrosus and nucleus pulposus tissues has been investigated in various research approaches [20,21]. Collagen, as a major component of a healthy IVD ECM, demonstrates biodegradability, poor antigenicity, and superior biocompatibility, and can be used for biomedical purposes, in the form of films, sponges, nonwovens, hydrogels, and micro- or nanoparticles [22,23,24].

The objective of this work is to examine injectable and cell-based hydrogel implants obtained from electrospun collagen nanofibers for their clinical use as a regenerative therapy within the context of the intervertebral disc nucleus in an ovine model. Self-assembling peptide nanofiber scaffolds, which are a new biological material with unique properties for cells, provide a new option in disc implant development. The nanofibers form hydrogels (>99.5% water) with structural and biomechanical properties similar to soft tissues and offer an optimal three-dimensional environment for cell growth and differentiation for various applications in regenerative medicine [25].

## 2. Results

### 2.1. Physiological Compatibility of Collagen Hydrogel

Laser scanning microscopy was used to investigate the quality of the developed collagen hydrogel system, composed of electrospun collagen fibers serving as the matrix for adipose stem cell growth. As seen in Figure 1, 2 h after their introduction into the hydrogel, the cells have already spread. After 36 h of incubation, the cytoskeleton is clearly formed (red) and the nuclei distinctly visible (blue). This indicates good cell vitality. As such, the physiological conditions necessary for supporting the ASCs over this time scale can be met by the hydrogel system.

### 2.2. Disc Height Analysis

Disc height analysis is based on a nine-point distance measurement in the anterior (A), central (C), and posterior (P) line of the discs with data collection at the left, middle, and right points. To ensure optimal data volume for statistical analysis, the anterior, central, and posterior measurements were averaged for each disc. All data sets are normally distributed. For the 6-month group, averaged anterior disc heights were 3.16 ± 0.48 mm for native discs, 2.78 ± 0.64 mm for damaged discs, and 2.75 ± 0.48 mm for collagen-treated discs. One-way ANOVA showed that these groups differ significantly (A; *p* = 5.4×10^−4^) (Figure 2). Averaged central disc heights for the 6-month group were 3.60 ± 0.56 mm for native discs, 3.01 ± 0.67 mm for damaged discs, and 3.18 ± 0.48 mm for collagen-treated discs, also showing significance (C; *p* = 5.4×10^−4^). Lower values were recorded for posterior disc height of the 6-month group: 2.35 ± 0.70 mm for native discs, 1.86 ± 0.84 mm for damaged discs, and 1.91 ± 0.66 mm for collagen-treated discs, with demonstrated significance (P, *p* = 0.0049). For the 12-month group, disc height data were as follows: anterior: 2.91 ± 0.46 mm native, 2.47 ± 0.43 mm damaged, 2.65 ± 0.52 mm collagen-treated; central: 3.70 ± 0.58 mm native, 2.87 ± 0.59 mm damaged, 3.19 ± 0.65 mm collagen-treated; posterior: 2.31 ± 0.72 mm native, 1.76 ± 0.48 mm damaged, 1.81 ± 0.66 mm collagen-treated. There are significant differences in the averaged values of all data sets (A; *p* = 8.2 ×10^−4^/ C; *p* = 4.2×10^−7^ /P; *p* = 4.6×10^−4^).

Beside variations in the disc heights, no morphological changes of the cartilaginous end plates could be detected in μ-CT examinations in either group (6-month or 12-month).

### 2.3. Histology

Figure 3 provides an exemplary comparison of the IVD histological investigations, including an enlarged view of the NP area of a native disc (Figure 3A,D), damaged disc (Figure 3B,E), and disc treated by injection of the ASC-loaded collagen hydrogel implant all at 6 months post-op (Figure 3C,F). When comparing the total cross sections of the discs, there is an obvious loss of height in the damaged IVD. Furthermore, the NP structure is clearly impaired as the cells are oriented along the lamellar structure and show increased formation of chondrones (see Figure 3E). The NP tissue of the native and treated discs appear unstructured, with uniform cell distribution. Although clear disruption of the disc structure is visible in the damaged disc shown in Figure 3B,E, especially within the NP area, the injection channel within the annulus fibrosus tissue is closed by scar-like tissue formation (white arrow). Furthermore, the NP structure is clearly impaired, as the cells are oriented along the lamellar structure and show increased formation of chondrones (see Figure 3E, enlargement).

Figure 4 depicts exemplary IVD samples from the 12-month group, harvested 1 year after treatment. Similar to the 6-month group, there is a loss of disc height in the degenerated disc in (Figure 4B), but the NP tissue (Figure 4C) does not show any structural irregularities the tissue matrix appears more compact. In comparison, the histology of the native IVD and collagen hydrogel treated IVD demonstrate normal disc height, cell density, and cell distribution within the tissue matrix (Figure 4A,C,D,F). The pronounced curvature of the vertebral end plates clearly visible in Figure 4C is an age-related characteristic of normal disc degeneration.

Using the prepared histological samples, disc degeneration was graded according to the methodology of Hoogendoorn [26]. Degenerative changes within the NP, AF, and the interface between them were assessed and subsequently compared. Results are shown in Figure 5. For the 6-month group, disc degeneration in the native disc tissue received a mean histological score of 2.56 ± 1.00, in damaged disc tissue a score of 4.33 ± 1.22, and in collagen-treated disc tissue a score of 3.67 ± 0.71. The mean histological scores within the 12-month group were: 2.89 ± 1.05 for native disc tissue, 4.11 ± 1.05 for damaged disc tissue, and 4.00 ± 1.00 for collagen-treated disc tissue. In both groups, the mean values of the native, damaged and collagen-treated discs were significantly different (*p* < 0.05).

## 3. Discussion

In this 1-year animal study, intervertebral disc degeneration was induced via minimally invasive partial nucleotomy of ovine IVDs followed by injection of a collagen hydrogel implant loaded with in vitro cultured autologous adipose stem cells.

The ovine model is particularly advantageous compared to other small and large animal models regarding its comparability to the clinical condition in humans for degenerative disc disease. As seen in humans, notochordal cells are no longer present in the NP of adult sheep, which means that tissue-induced self-regeneration can be excluded [27]. Furthermore, the anatomical conditions and biomechanical stress levels of sheep correspond approximately to those of humans [28]. Due to the possibility of subsequent open grazing and a demonstrated tolerance to surgical interventions, the sheep animal model is, from an ethical point of view, established in the research and development of disc degeneration and regeneration approaches.

The animal model used [29] can effectively depict traumatic situations and is suitable for conducting long-term studies of up to one year. For the examination of cell-loaded scaffolds, however, the model described in this study was modified so that discography using an X-ray contrast medium was not necessary. Contrast discography requires an additional puncture of the disc to inject the iodine-based X-ray contrast medium, and the effects if may have on the cell-loaded scaffolds is not explicitly known and would require additional testing. Keeping clinical practice in mind, however, this is not necessary, as discography is not performed prior to a regenerative therapy.

In disc herniation, a traumatic event leads to rupture of the AF and leakage of NP tissue, which is associated with a decrease in IVD height. Within 6 weeks after herniation, pressure-stable scar tissue forms over the ruptured area of the AF [29]. In application, a constantly increasing counter pressure builds up in the injection syringe in the case of a pressure-tight AF closure by scar tissue when the cell-loaded scaffold is injected. As this was seen in all animals in the present study, it can be concluded that all regenerative-treated intervertebral discs had an intact AF on the day of injection. None of the animals showed neurological deficits, indications of behavioral changes regarding their natural inclination for movement, or a reduced physiological range of motion over the course of the experiment. Therefore, it can be concluded that the application of the cell-loaded scaffold was successful in all 18 discs.

There is currently a scientific consensus that only cell-loaded scaffolds can stop or slow further progression of IVD degeneration after disc herniation. Due to the lack of notochordal cells in human adult disc tissue, there is no cell proliferation within the disc and only limited matrix production of the existing cells [30]. Cells can migrate into the NP via the endplate, but this must be stimulated, which is generally achieved by mechanical stimulation [11]. On this basis, cell-free systems can only stabilize the disc for a certain of amount of time. The degeneration cascade of the IVD pauses until the material degeneration begins. At this point, the cascade picks up again, so a desired long-term effect is not achieved.

The known small animal models (rat and rabbit) usually distort the potential success of materials, because in these animal models notochordal cells are present, which favor regeneration. This is not the case in sheep and goat [31,32,33] as well as dog [34,35] animal models. These models reflect a more representative and realistic situation for humans and show that this combination of cells and material is a promising approach to minimize degeneration. In combination with a collagen-based scaffold material this has already achieved clinical success [3,36].

A further challenge for the successful establishment of a regenerative therapy approach is the identification of suitable cells and their source, which are used in combination with the scaffold for the purpose of disc regeneration [37]. As has been shown, the use of therapeutic approaches based purely on autologous chondrocytes [38] was not convincing despite initial success [39]. In particular, depending on the patient‘s age intervertebral disc degeneration (IDD) can be seen prior to disc herniation [40,41]. This issue indicates the need for an alternative autologous cell source. Therefore, MSCs or ASCs have been suggested in a number of contemporary studies. Both cell types are able to differentiate towards chondrocytes [11,42,43,44,45,46]. Due to a lower expected complication rate in tissue harvesting and a higher patient acceptance, focus has been placed on the use of ASCs. Using ASC as a regenerative approach for articular cartilage as well as for intervertebral disc tissue has been previously described [47]. In addition to basic research, ASCs have already been successfully used in studies of disc regeneration in large animal models such as dogs and goats [31,32,33]. With the described processing method, >5 × 10^7^ cells per animal could be cultivated from a few grams of sequestered adipose tissue (average approx. 10 g) after 3 passages. In fact, it took much less time than the available 6 weeks between surgeries; as such, the cells were cryopreserved until injection preparations were made. The available cell count in all animals was higher than the cell count proposed for ADCT therapy (1 × 10^6^ cells/disc) [31]. However, the ADCT cell source is not always available or the cells are in an advanced stage of degeneration [8] so that the technique of autologous in vitro cultivation is no longer successful. With approximately 2.5 × 10^6^ cells/disc, the applied cell count falls within the same range as other studies [31,48]. The natural cell density in human nucleus pulposus tissue is in a range of 3000 to 5000 cells/mm³ [49,50]. The number of cells used (5500 cells/mm³) was intentionally chosen to be slightly higher, since differentiation of the ASC’s by co-culture is intended in the animal model. In vitro studies demonstrate that ASCs begin to differentiate in the presence of chondrocytes [51,52,53,54] and this potential was used in the present study. By damaging the IVD via partial nucleotomy, there are still a sufficient number of chondrocytes within the remaining NP tissue available to induce differentiation. However, it must be noted as a limitation of the model that, contrary to the typical clinical situation, it cannot be assumed that an advanced degeneration cascade has developed at the cellular level within 6 weeks after partial nucleotomy, as is the case after a prolonged degeneration process prior to a human disc herniation. This means that in this model the ASCs are in co-culture with intact chondrocytes in situ following their injection. It can be assumed that there is an almost uniform stimulation of the ASCs. In the marginal area of the injected hydrogel and native NP tissue, a diffusion of messenger signals can be assumed, comparable to in vitro co-culture on a porous membrane. Free diffusion, limited by the thickness of the hydrogel body, is accompanied by forced convection from permanent cyclic compression and relaxation within the disc, as experimental [51,54] and numerical calculations have demonstrated [52,53,55]. For the described animal model, it can be assumed that convective mass transport dominates within the IVD, given the grazing husbandry used for this study and sheep’s natural predilection for movement in pasture. As cell culture studies show (Figure 3), there is no significant decrease in cell count over the test period of 36 h. This indicates that the purely physical cross-linking of the hydrogel body leads to advantageous diffusion properties. The viscosity of the hydrogel has been adjusted so that it does not escape through the puncture channel. This is also assisted by the needle tip geometry of the 21 G × 120 mm SPROTTE® cannula (Pajunk, Geisingen, Germany) selected in this study. The tip geometry and the lateral outlet ensure that no tissue is punched out during the puncture. Since the individual fibers of the AF are still relatively elastic in a 2-year-old sheep, the fiber composite is spread by the syringe and closes again after the needle is removed.

Radiological examinations of the treated motion segments showed no further visible degeneration features. Analysis of the disc height suggests that in some of the animals, degeneration continues despite treatment. However, the individual results of the height analysis also show that a good stabilization of the IVD height was achieved in a subset of the group of the treated IVD.

The Hoogendoorn score [26] also provides comparative results. This histological score was originally developed for a goat animal model and evaluates intervertebral disc degeneration based on the degeneration of the AF, the NP, and the AF/NP interface. For each tissue area, a score between 0 (normal) and 2 (degenerated) can be assigned. For the damaged, non-treated IVDs and the IVDs treated with the cell-loaded collagen hydrogel, the mean Hoogendoorn scores show that no significant differences can be detected. This is true for the animals of the 6-month group as well as the 12-month group. In the 6-month group, however, the range of variation is significantly larger in the undamaged discs than in the treated ones. For the animals of the 12-month group, the range of variation is comparable. The Hoogendoorn scores of the damaged, non-treated IVDs in both groups are comparable to the values described in Schwan et al. [29].

## 4. Materials and Methods

### 4.1. Animal Model

This study received official approval by the Leipzig National Directorate (experiment No. TVV 30/11). It includes data from 18 Merino sheep, all healthy adult females of mean weight 65.39 ± 7.88 kg and 2 years of age at the time of surgery. Test animals were evaluated for general health and mobility to preclude pre-existing diseases or musculoskeletal deficits. The animal model (see Figure 6) is designed in 3 steps, which first describes a minimally invasive injury to the ovine lumbar discs via partial nucleotomy to induce intervertebral disc degeneration [28]. In the second step, degenerated discs were treated by injection of a collagen hydrogel scaffold loaded with autologous adipose stem cells in a minimally invasive surgery. The 18 sheep were divided into 2 experimental groups of 9 animals each. After recovery times of 6 and 12 months, respectively, each group of animals was euthanized, and their native, degenerated, and regenerated discs excised for µ-CT and histological investigations.

### 4.2. Cell Isolation and Cultivation

During the disc injury surgery, a few grams of subcutaneous fat tissue was extracted from each sheep from the sacral region to isolate autologous adipose stem cells. For preparation, the harvested adipose tissue was rinsed in sterile Dulbecco’s PBS (c-c-pro, Oberdorla, Germany; supplemented with 1% penicillin/streptomycin, c-c-pro, Oberdorla, Germany; 1% amphotericin B; c-c-pro, Oberdorla, Germany; pH 7.4), freed from excess connective tissue and, subsequently, minced into small pieces.

Cell isolation from adipose tissue was performed immediately after tissue extraction by enzymatic dissociation using Collagenase Type I solution (Worthington, Lakewood, NJ, USA, 50 U/mL in DMEM, c-c-pro, Oberdorla, Germany plus 1 mL BSA stem solution (10 mg/mL), Sigma-Aldrich, Schnelldorf, Germany). For dissociation, tissue was incubated in 25 mL of enzyme solution for 1h at 37 °C with slight shaking and then pressed through a sieve and rinsed with Krebs–Ringer solution. This was followed by centrifugation of the solution for 10 min at 300 *g* at room temperature. After the supernatant was removed, the resulting pellet was resuspended in PBS and centrifuged again. Finally, the cell pellet was resuspended in DMEM (supplemented with 10% autologous sheep serum; 1% L-glutamine, c-c-pro, Oberdorla, Germany; 1% penicillin/streptomycin; and 1% amphotericin B) and transferred to a T75 cell culture flask.

In parallel to enzymatic dissociation, a few tissue pieces are placed into a sterile petri dish immediately after mincing and floated with supplemented DMEM. This allows the cells to grow out of the tissue during incubation. The remaining tissue pieces were removed from the dishes after 48 h of incubation. Both approaches were incubated at 37 °C and 5% CO_2_ until 80% confluency and further sub-cultivated to expansion, up to a cell number sufficient for preparing cell-loaded implants for four intervertebral discs for each sheep. It is possible to expand the cells to a total cell number of 5 × 10^7^ cells per animal with three passages and to then cryologically store them until the implantation procedure. Cell morphology was observed by light microscopy. For preservation, the cells were removed from the growth surface using Trypsin-EDTA (c-c-pro, Oberdorla, Germany), centrifuged at 300 *g* for 5 min, taken up in cryo-preservation medium (cryo safe I, c-c-pro, Oberdorla, Germany), and stepwise frozen.

Three days prior to implantation, the cells were thawed and cultured up to the day of implantation to ensure active cell metabolism at the time of injection. Cell culture was arranged under standard conditions and using standard cell culture media with the addition of autologous serum obtained from each animal individually. The sheep serum was produced out of fresh blood, which was taken from each animal during anesthesia directly before the disc injury surgery.

### 4.3. Collagen Hydrogel Fabrication

The collagen hydrogel used in the cell-loaded disc implants for injection into the nucleus pulposus area of degenerative intervertebral discs was produced from electrospun bovine type I and II collagen (SpinPlant GmbH, Halle, Germany) with fiber diameters between 50–300 nm. Prior to hydrogel preparation, collagen nonwovens were sterilized via UV irradiation for 2 h and cut into 1 mg portions. Each 1 mg of collagen nonwoven was subsequently mixed into 500 µl of a 1:1 solution of PBS and autologous heat-inactivated serum to form an injectable viscous hydrogel. The hydrogel was stored ready to use in sterile high-pressure Medaillon^®^ syringes until the injection surgery.

### 4.4. Physiological Compatibility of Collagen Hydrogel

To confirm the physiological compatibility of the collagen hydrogel system, cell behavior was qualitatively investigated by immunostaining and subsequent laser scanning microscopy in previous experiments. For viewing, the ASC loaded collagen hydrogel was loaded onto coverslips. After an incubation time of 30 min, expansion media (ThermoFisher Scientific, Dreieich, Germany) was added followed by a second incubation period of 2 h and 36 h. In the next step, the cells were fixed with 4% paraformaldehyde (ThermoFisher Scientific, Dreieich, Germany) and immunostained with Rhodamine Phalloidin marked antibody (ThermoFisher Scientific, Dreieich, Germany) that selectively binds F-actin to reveal the cytoskeleton and subsequent DAPI (Sigma-Aldrich, Schnelldorf, Germany) staining of cell nuclei.

### 4.5. Injection of Cell Loaded Collagen Hydrogel into IVDs

As the second component for the hydrogel, 500 µL of cell solution (5.5 × 10^4^ ASC in PBS serum solution, 1:1) was prepared from autologous cell culture of adipose stem cells. The cell solution was also kept ready to use in syringes until the time of the injection surgery.

Before hydrogel injection, each intervertebral disc to be treated was visualized via C-arm X-ray and anatomical references were marked on the shorn skin of the animal. Meanwhile, the cell suspension was injected homogeneously into the syringe containing the collagen hydrogel. By use of a lumbar puncture cannula and the application of continuous, steady pressure, a portion of the cell-loaded hydrogel implant was injected within the NP region of the disc. Images of the injection are seen in Figure 7.

### 4.6. Postoperative Protocol

Each test animal was monitored for 1 week in an indoor recovery enclosure following surgery. Afterward, all animals were returned to a standard livestock farm and released for daytime grazing and fully unrestricted movement.

All animals were euthanized in accordance with their respective group designation at 6 or 12 months after surgery. The lumbar spine of each animal was fully excised, cleaned of residual soft tissue, and frozen at −20 °C. The spines were then dissected to isolate the motion segments. All segments were refrozen at −10 °C and then deep-frozen to −80 °C for 24 h before µ-CT analysis. Samples were stored at −36 °C to prevent postmortem manipulative tissue damage or degeneration.

### 4.7. µ-CT Analysis

μ-CT imaging enabled non-destructive, three-dimensional characterization of tissue samples with high spatial resolution, providing information on disc volume and the behavior of the disc-vertebra interface. μ-CT was performed using a phoenix nanome|x 180NF (GE, USA), an X-ray cone-beam system with a flat detector array and image amplifier, that operated with increasing radiation intensity and extended integration times. Eleven single images with a 400 ms exposure time were collated by the CCD detector for each of the 500-image pattern steps over a 360° rotation. Images of exceptional quality, used for calculating 3D volumes, were produced using 100 kV acceleration voltages and 100 μA beam current. The resulting voxel size was 25 microns, allowing visualization of the IVD lamellar structure. Average disc height was evaluated along the disc’s anteroposterior midline from μ-CT radiographs [56] as shown in Figure 8.

### 4.8. Histology and Light Microscopy

Motion segments were fixed for 7 days in 2.5 % glutaraldehyde (Carl-Roth, Karlsruhe, Germany) PBS buffer, re-frozen, and trisected. Samples were dehydrated via an ascending ethanol (Carl-Roth, Karlsruhe, Germany) series and embedded in Technovit9100 resin (Heraeus Kulzer, Germany). Resin blocks were cut into 2 mm thick slices, fixed to microscope slides, and polished to 50 μm. Samples were Masson–Goldner (Carl-Roth, Karlsruhe, Germany) stained [57], and then visually recorded using Slide Scanner ZEISS Axio Scan.Z1 (Zeiss, Germany). Histological grading was conducted according to the histological grading scale for goat IVD degeneration proposed by Hoogendoorn et al. [26], which included the grading of the pattern of AF, the border between the AF and NP, as well as the NP matrix itself from 0 (normal disc) to 2 (degenerated disc). Therefore, the maximum degeneration score is 6, which represents a severely degenerated disc.

### 4.9. Statistics

Both descriptive statistics and analysis of significance were performed with the statistical software OriginLab (OriginLab Cooperation, Northampton, MA, USA). Distribution analyses was done using the Kolmogorov–Smirnoff test with *p* = 0.05. Analysis of significances were examined by one-way ANOVA Tukey test with a confidence interval of 95 % (significance level *p* > 0.05).

## 5. Conclusions

This study addresses two of the five areas identified by Gullebrand et al. that are necessary for the development of a translational regeneration model and should therefore be considered in the design of future studies [37]. On the one hand, the sheep animal model is a clinical model at a relevant scale in terms of average disc size compared to humans. On the other hand, the use of subcutaneous adipose tissue as a cell source proves to be a promising approach as already discussed. As an outlook of this study, the addition of a translational preconditioning of the ADSC, as well as an extended experimental evaluation, can be mentioned. This should include the acquisition of biochemical markers, the comparison of the mechanical properties of degenerated and regenerated motion segments, as well as the development and integration of high-resolution evaluation methods based on radiological imaging techniques (e.g., MRI) to ensure a broader spectrum for the correlation of study results of different approaches.

## Figures and Tables

**Figure 1 ijms-22-04248-f001:**
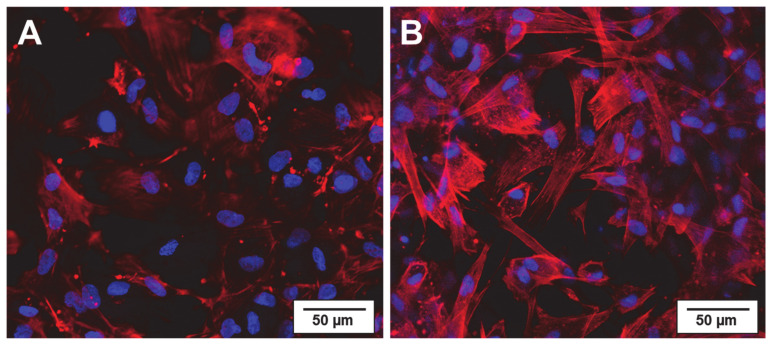
Light scanning microscopy of ASCs in the collagen gel system (**A**) after 2 h of incubation, (**B**) after 36 h of incubation. Immunostaining of cytoskeleton with F-actin marked antibody and fluorescent staining of cell nuclei with DAPI.

**Figure 2 ijms-22-04248-f002:**
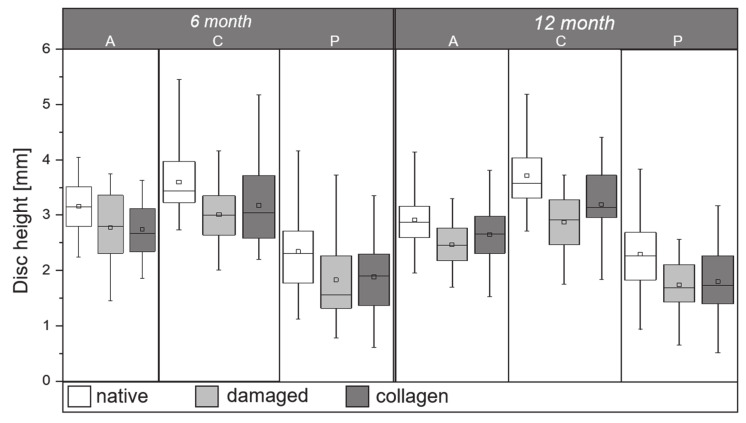
Boxplot comparing the distribution of anterior (**A**), central (**C**), and posterior (**P**) disc height measurements between the native (white), damaged (light gray), and ASC-loaded collagen hydrogel treated (dark gray) segments at 6 and 12 months after treatment. The threshold for significance is *p* < 0.05.

**Figure 3 ijms-22-04248-f003:**
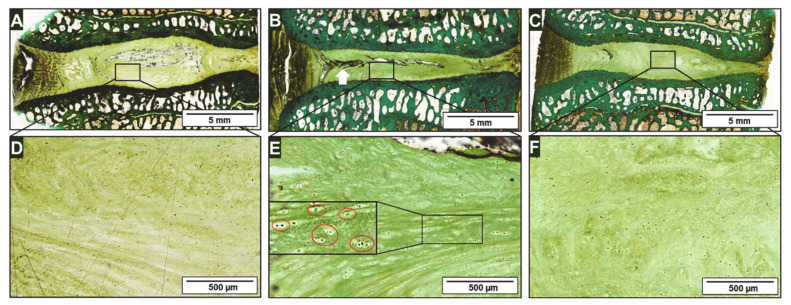
Light microscopy of Masson-Goldner histology of a representative (**A**) native disc segment, (**B**) damaged segment, and (**C**) cell-loaded collagen hydrogel injected segment at 6-months post-op. (**D**–**F**) Enlarged views of the nucleus pulposus area are shown.

**Figure 4 ijms-22-04248-f004:**
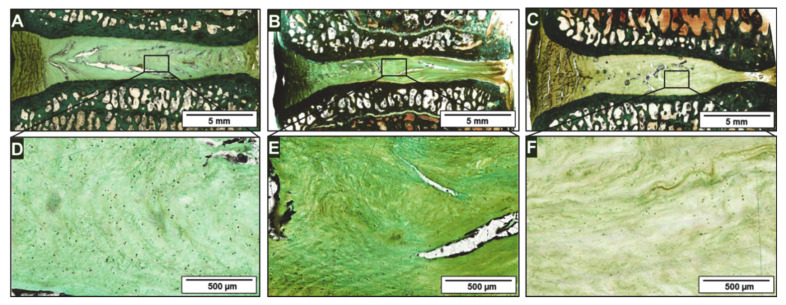
Light microscopy of Masson-Goldner histology of a representative (**A**) native disc segment, (**B**) damaged segment, and (**C**) cell-loaded collagen hydrogel injected segment at 12-months post-op. Enlarged views of the nucleus pulposus area are shown (**D**–**F**).

**Figure 5 ijms-22-04248-f005:**
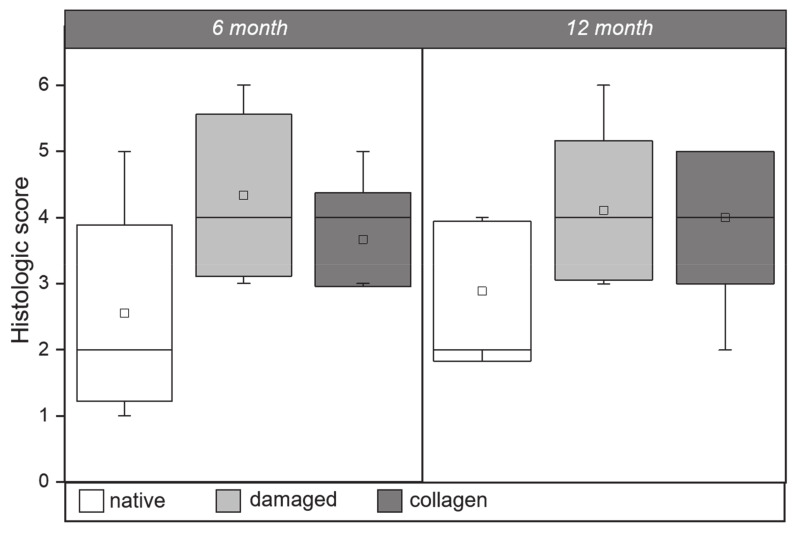
Average histological scores of native (white), damaged (light gray), and ASC-loaded collagen hydrogel treated (dark gray) disc segments at 6 and 12 months after treatment. The threshold for significance is *p* < 0.05.

**Figure 6 ijms-22-04248-f006:**
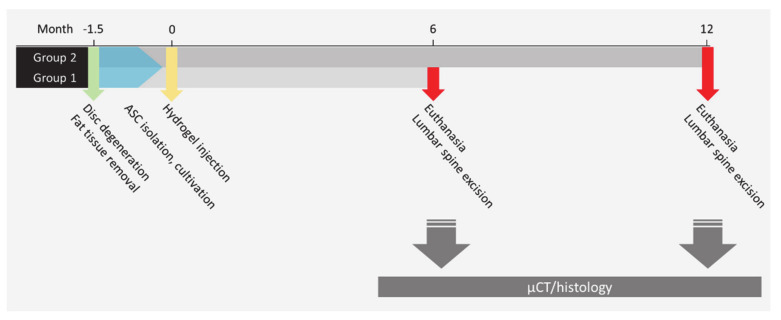
Scheme of the ovine animal model for intervertebral disc regeneration.

**Figure 7 ijms-22-04248-f007:**
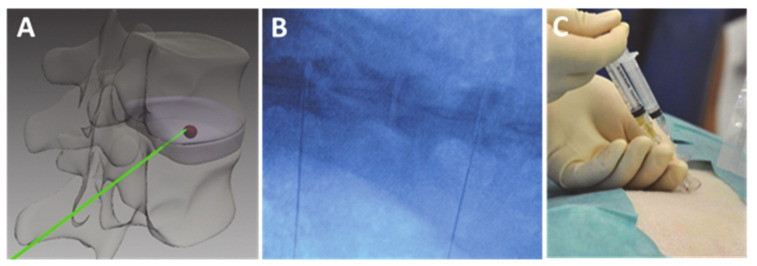
(**A**) Schematic illustration of intervertebral disc injection; (**B**) X-ray image of the ovine lumbar spine with lumbar puncture cannula within the IVDs during injection procedure; (**C**) photographic image of the injection.

**Figure 8 ijms-22-04248-f008:**
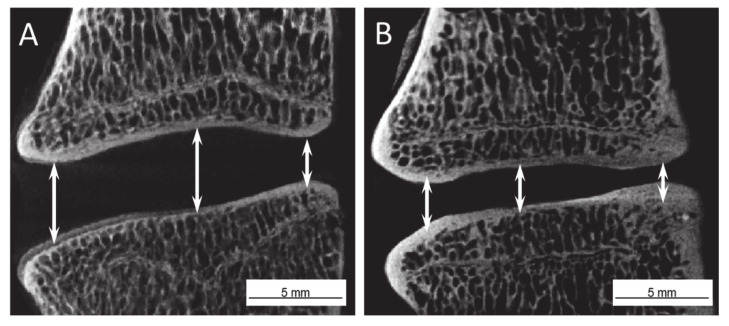
Representative images of the intervertebral disc height measurement at 3 different measurement points in the central cross-section plane: (**A**) control segment without partial nucleotomy, (**B**) severely damage after partial nucleotomy without treatment.

## Data Availability

Not applicable.

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
