# Peer review of "Intervertebral Disc Regeneration Injection of a Cell-Loaded Collagen Hydrogel in a Sheep Model"

_ijms, 2021, doi:10.3390/ijms22084248_

Round 1
Reviewer 1 Report
Intervertebral disc (IVD) degeneration is a frequent cause of back and neck pain, which are major public health concerns in modern society. Limitations of current pharmacological treatments and invasive surgical procedures prompt the development of new therapeutic options.
The current manuscript evaluated the replacement of a degenerated disc with a collagen scaffold containing reparative cells, namely autologous adipose tissue-derived stem cells, as a promising strategy for IVD repair. Of significance, a large animal model (sheep) with induced disc degeneration was used, which increased the translation potential of engineered construct toward clinical use in human patients. Therefore, this is a topical and relevant manuscript to the treatment of IVD degeneration.
This research presents an interesting topic but some concerns need to be cleared.
Results:
Autologous adipose stem cells (ASC) isolation and viability in collagen hydrogel:
- How the authors guarantee that cells isolated from autologous adipose tissue met the requirements of mesenchymal stem cells (MSC)? For example, do you analyze the osteogenic, adipogenic, or chondrogenic differentiation capacities of these cells? Or if the cells were positive for CD90 and CD29 (MSC markers) and negative for CD45 and CD34 (hematopoietic markers)?
- Since or in vivo experiments lasts up to 12 months, do you investigate ASC viability in collagen hydrogel for longer periods than 36h? What about ASC differentiation in collagen hydrogel?
Disc height and histology analysis:
- Do you perform control experiments with only collagen hydrogel, this means without ASC?
- The results on µCT examinations are missing
- The authors verified that native, damaged and collagen-treated discs differ significantly. However, statistical analysis of damaged vs control and collagen vs damaged are mandatory.
- Radiological examination was performed on a follow-up basis (examinations at different time points during the study) or only at the endpoints (6 and 12 months)?
Discussion:
- In this section, the authors extensively discuss technical issues (animal model, Hoogendoorn score, ...) with many assumptions on ASCs and hydrogel behavior in vivo. Consider shortening the discussion of these issues; instead, present an in-depth discussion of the obtained data. The clinical significance of presented results, challenges to overcome before clinical translation, and future studies should be highlighted. As reference, please see the following articles:
Ishiguro H et al. Acta Biomaterialia. 2019; 87: 118–129
Gullbrand SE et al. JOR Spine. 2018;1(2):e1015.
Clouet J et al. Advanced Drug Delivery Reviews. 2019; 146: 306–324
- In the in vivo environment, engineered construct face nutritional and mechanical challenges (hypoxic, acidic and nutrient-deprived microenvironment, exposition to high pro-inflammatory cytokines and reactive oxygen concentrations, as well as abnormal physical loads) that likely compromise ASC viability.
To what extend contribute cell-free collagen hydrogel to the observed results?
How ASC preconditioning or pre-differentiation before implantation into IVD could enhance cell survival and matrix formation?
Materials and Methods:
Please give further information on experimental procedures and/or provide the corresponding references. For example (but not limited), detail cell isolation and cultivation (enzymatic dissociation, standard culture conditions, cell culture media, thaw media,...), cell expansion media, etc (include commercial suppliers of reagents). Please remember that you should include adequate experimental data for others to be able to reproduce your work.
Conclusion:
Include future perspectives and recommendations.
Minor issues:
line96-99: A reference is missing
line 109-111: Please remove
line 155: please indicate chondrones formation in figure 3
line 159: white arrow indicating scar-like tissue formation in Figure 3, is missing
line291: please revise reference citing
Figures 2 and 5: "** not significant" seems useless and confusing.
Author Response
see file

Reviewer 2 Report
In this article, authors proposed a collagen-based hydrogel loaded with mesenchymal stem cell for intervertebral disk regeneration.
I recommended some revisions.
Comments:
- Materials and method section, page 8, line 291: there is an error in the reference.
- Results section, page 3, line 109: the first part is only an indication of the editor to explain how divide the section.
- Cells wer loaded on the scaffold and not in the scaffold. Adding the cells into the hydrogel it is possible to prolong the cell release. Why to you chose to evaluate only this type of mixture? Which are the advantages correlated to the use of this physical mixture?
- Figure 2. Thresholds indicated only the difference between native and collagen groups. Which are the differences between damaged and collagen groups? Could be important to evaluate if the treatment had an effect of all measured parameters.
- Overall, no differences were substantially founded between damaged and treated animals. Do you think to modify the scaffold type, the cell loading method or other factors to improve the efficacy of the therapy?
- No information about further studies were presented.
Author Response
See the files responses (1x) and revised manuscript (1x)

Round 2
Reviewer 1 Report
Minor revisions:
Line 255: Define ADCT
Line 258: "In vitro" in italics
Line 437: Define ADSC
Reviewer 2 Report
Authors modified the manuscript as requested.